# Does the Tone in Corporate Social Responsibility Reports Misdirect Analysts' Forecasts in China?

Xiaoying Liang and Hongjun Wu *

School of Management, Xiamen University, Xiamen 361005, China
* Correspondence: wuhongjun@xmu.edu.cn

**Abstract:** With increasing emphasis being placed on corporate social responsibility, the number of companies furnishing corporate social responsibility (CSR) reports is increasing. This study investigates the impact of abnormal positive tone in CSR reports on analysts' earnings forecast bias. The textual analysis of CSR reports of Chinese listed companies between 2006 and 2016 reveals that an abnormal positive tone significantly and positively relates to an optimistic bias in analysts' forecasts. This effect is pronounced among companies with poor financial transparency and those operating in regions where culture is stakeholder-oriented. Further analysis confirms that the poorer the company's CSR performance, the more it tends to mislead analysts using an abnormal positive tone in its CSR report. Based on these findings, this study suggests that firms may greenwash using an abnormally positive tone in their CSR reports.

**Keywords:** tone management; analysts' optimistic bias; corporate social responsibility disclosure; financial transparency; stakeholder orientation

## 1. Introduction

This study examines whether the tone in corporate social responsibility (CSR) reports affects the earnings forecasts of analysts. With rapid economic development, stakeholders, such as the government and the public, are all increasingly focusing on companies' social and environmental behavior. Consequently, an increasing number of listed companies have voluntarily or compliantly furnished CSR reports.

Some studies suggest that CSR disclosure significantly influences capital markets [1–3]. By providing incremental information, it is the most direct way companies can demonstrate their CSR performance to the public and the market. Additionally, CSR disclosure supplements a company's financial disclosure, enabling a comprehensive view of the company's value. For instance, analysts emphasize and integrate the information from CSR reports into their financial forecasts [4–6]. Studies on Chinese capital markets show that capital markets can not only enhance their understanding of a company's operations through CSR reports but can also gauge its future profitability [7,8].

However, it is challenging to accurately quantify CSR performance owing to the broad scope of CSR. Therefore, the tone used in the CSR report may influence stakeholders' judgment of the firm's CSR activities and performance, thereby influencing the effects of CSR disclosure. Research on the textual tone used in corporate disclosures suggests that textual tone, a non-quantitative type of information, has a significant impact on the market and investors. A positive tone in corporate disclosures will increase the disclosure's persuasiveness, inducing investors to have a more optimistic value judgment [9–12]. Therefore, companies are motivated to manage the tone when disclosing their information [13].

Currently, there are no clear regulations regarding the content and language of CSR disclosures in China, and listed companies hold a great degree of discretion over the language used in the reports. Therefore, it is important to determine whether companies use abnormally positive language to form a favorable image and manipulate the judgement of capital markets. However, research on this topic is scant.

Tone management in the text is the use of an abnormal tone besides a neutral tone [12]. Chinese is a unique language, and this study builds a Chinese tone dictionary list with CSR characteristics to identify whether and how tone management, particularly the use of an abnormally positive tone in CSR reports, misleads judgment in the capital markets and induces a bias in analysts' forecasts. Studies have shown that firms' financial transparency and regional culture moderate the relationship between CSR disclosure and the accuracy of analyst forecasts [5]. Therefore, we examine the moderating effects of these two factors. The results reveal that the relationship between abnormal positive tone in CSR reports and analysts' earnings forecast bias is strong when a company's actual CSR performance is poor.

Research on whether companies use abnormally positive language to form a favorable image and manipulate capital markets' judgments on their CSR performance is scant. We use a self-constructed CSR tone Chinese dictionary to demonstrate the serious consequences of misinforming analysts using an abnormally positive tone in CSR disclosures in China. We contribute to the existing literature in the following ways. First, our paper complements the growing literature on the economic consequences of textual tone in CSR reports. Few studies have explored how the textual tone in CSR reports influences capital markets, particularly from the perspective of greenwashing. Many studies have discussed how capital markets are affected by incremental quantitative information in CSR disclosures, which are the most important form of non-financial disclosure and provide information on a company's capital costs, their value, and the efficiency of their information dissemination [1,11]. However, previous studies have paid little attention to the non-quantitative information in CSR disclosures. Muslu, Mutlu, Radhakrishnan, and Tsang [14] studied the financial tone of the narratives of CSR disclosures and found that a certain tone caused more accurate forecasts; however, they did not distinguish between normal and abnormal tones. Focusing on non-quantitative information, specifically the textual tone in CSR reports, we find that an abnormally positive tone in CSR reports will bias the judgment of analysts, thus expanding the body of knowledge on social responsibility.

Next, this study proves that CSR disclosures misdirect market judgment. Existing studies have primarily focused on whether CSR reports provide effective incremental information and their impact on companies internally and externally. However, they do not explore the mechanism behind their impact. This study extracts the abnormally positive part, containing "positive reversal," from the tone in CSR reports to analyze the causal relationship between the abnormally positive tone and bias in analysts' forecasts. Thus, it provides empirical evidence on how tone in CSR reports misdirects analysts' judgment.

Finally, examining the misdirection induced by the tone used in CSR reports and the related mechanism enhances stakeholders' understanding of CSR reporting. China's emerging capital markets are yet to mature, and there is still room to improve laws and regulations. While CSR disclosures can provide incremental information, its non-quantitative information may be used to misdirect the market, reducing the operational efficiency of capital markets. Hence, this study has salient policy implications.

The remainder of this paper proceeds as follows. Section 2 presents the hypotheses. Section 3 discusses the research methodology, while Section 4 reports the empirical results. Section 5 discusses the tests of robustness. Section 6 explains the potential mechanism behind the impact of tone management on analysts' forecast bias. Section 7 concludes.

## 2. Hypotheses Development

### 2.1. Abnormally Positive Tone in CSR Reports and Analysts' Earnings Forecast Bias

According to the instrumental stakeholder theory, there are tangible financial benefits associated with providing CSR reports [4]; however, it is challenging to quantify and assess CSR performance as the reports contain a great degree of soft disclosure. Additionally, firms have a lot of discretion over the content of the CSR reports, allowing them the opportunity to manage the tone and potentially mislead analysts' forecasts.

Studies find that emotional tendencies in the tone of disclosures can provide incremental information regarding a company's future earnings and complement the quantitative information [11,15]. Since a highly positive tone can increase the persuasiveness of textual information, analysts can extract valuable information and adjust forecasts accordingly [11,12]. Moreover, managers may employ a high-spirited tone while presenting their firm's performance to take advantage of the limited attention and subjective prejudice of investors, and disseminate or conceal certain fundamentals of the company, thus engaging in impression management [9,10,15–18].

Due to the absence of a unified standard in China, the usage, credibility, and assessment of the tone used in CSR disclosures are highly debatable. Companies generally use a high-spirited tone in their CSR report to exhibit socially responsible behavior and performance [12]. Therefore, the question arises: does this abnormally positive tone influence the external stakeholders, including analysts, the public, and other companies? Among those stakeholders, it is of particular importance to determine whether it misleads analysts, considering that analyst forecasting significantly influences investors' judgments and beliefs [5].

This study characterizes textual tone into two types: normal and abnormal tone. The former aligns with the company's actual performance, while the latter represents its tone management behavior. An abnormally positive tone reflects the non-essential components in tone descriptions based on the residual value obtained from a tone model that controls for factors such as company performance and risk, thus quantifying possible tone management behavior [12]. Since a normal tone reflects a company's quantitative information, and an abnormally positive tone represents its tone management, we propose the following hypothesis:

**Hypothesis 1 (H1).** *An abnormally positive tone in CSR reports is positively related to analysts' earnings forecast bias.*

### 2.2. Moderating Effect of Firm's Financial Transparency

Financial transparency improves the information environment, increases market liquidity, lowers a company's cost of external financing, and reduces fluctuations in its value and valuation errors [16,19–21]. Meanwhile, maintaining information transparency increases the impact of external supervision on corporate behavior. When a company's financial transparency is poor, CSR reports become a crucial source of private information, other than the company's financial disclosures, which can provide incremental information at lower costs [5,22–24]. As CSR reports can complement financial disclosure, when both financial transparency and disclosure quality are poor, analysts must rely on other information sources. At that point, the tone in CSR reports, which is a carrier of private information and is more useful for firms with greater financial opacity, will have a significant effect on analysts' earnings forecasts. Hence, we propose the following hypothesis:

**Hypothesis 2 (H2).** *The poorer the firm's financial transparency, the stronger the relationship between abnormally positive tone in CSR reports and analysts' earnings forecast bias.*

### 2.3. Moderating Effect of Regional Stakeholder-Oriented Culture

According to the resource dependence theory, a firm's profitability depends on the resources present in its business environment. When the regional culture is stakeholder-centric, firms must verify that they are socially responsible in order to obtain legitimacy. A firms' CSR performance will have significant effects on stakeholders' attitudes and support, which determines the performance and future earnings of firms [5]. Thus, if firms in these regions intend to receive recognition from stakeholders, they may use an outstanding and eye-catching tone in their CSR reports. At the same time, tone plays a crucial role in analysts' subjective estimation of the financial performance of firms in such regions, as CSR performance and its disclosure are associated with financial benefits [4]. Thus, they may be

significantly affected by an abnormally positive tone in CSR reports, creating a bias in their earnings forecasts. Accordingly, we propose the following hypothesis:

**Hypothesis 3 (H3).** *The higher the level of regional stakeholder-oriented culture, the stronger the relationship between abnormally positive tone in CSR reports and analysts' earnings forecast bias.*

## 3. Research Design

### 3.1. Definition of Variables

3.1.1. Analyst Earnings Forecast Bias (*BIAS1*)

We define analyst earnings forecast bias as the difference between analysts' earnings forecasts and the actual value of a company's earnings divided by the company's average opening price at the beginning of the year [9]. Equation (1) presents the formula of calculating this variable.

$$BIAS1_{i,t,j} = \frac{Feps_{i,t,j} - Aeps_{i,t}}{P_{i,t}}, \tag{1}$$

where subscripts $i$, $t$, and $j$ denote firm $i$, year $t$, and forecast $j$, respectively. $Feps_{i,t,j}$ is the analysts' forecast of the company's earnings per share (EPS) for the current year (year $t$) and on the day following the publication of the company's CSR report. $Aeps_{i,t}$ is the actual EPS for the current year, and $P_{i,t}$ is the company's opening price for the current year.

3.1.2. CSR Performance (*KLD*)

The Chinese Research Data Services (CNRDS) database provides the score of firms' CSR performances. In the CNRDS database, CSR is divided into six dimensions: "community, volunteer program, and social controversy," "environmental protection", "corporate governance", "employee relation", "diversity", and "product." Each dimension is scored from two perspectives: strengths and concerns. *KLD* is calculated by summing the score of all six dimensions.

3.1.3. Abnormally Positive Tone in CSR Reports (*ABTONE*)

Following Price, Doran, Peterson, and Bliss [10], Brockman and Cicon [11], and Huang, Teoh, and Zhang [12], the tone in CSR reports is calculated as follows:

$$TONE_{i,t} = \frac{POSPCT_{i,t} - NEGPCT_{i,t}}{POSPCT_{i,t} + NEGPCT_{i,t}}, \tag{2}$$

where *POSPCT* is the percentage of positive words and *NEGPCT* is the percentage of negative words out of the total number of words in a CSR report.

To distinguish between normal and abnormal tones in CSR reports, we define abnormally positive tone (*ABTONE*) as the residual term of regressing tone on the determining factors, following Huang, Teoh, and Zhang [12], and satisfying the following quantitative relationship:

$$\begin{aligned} TONE_{i,t} = \beta_0 + \beta_1 \cdot COM_{i,t} + \beta_2 \cdot EVN_{i,t} + \beta_3 \cdot CGOV_{i,t} + \beta_4 \cdot EMP_{i,t} + \beta_5 \\ \cdot DIV_{i,t} + \beta_6 \cdot PRO_{i,t} + \beta_7 \cdot LISTAGE_{i,t} + \beta_8 \cdot SIZE_{i,t} + \beta_9 \\ \cdot ROA_{i,t} + \text{YEAR FE} + \text{INDUSTRY FE} + \varepsilon_{i,t}, \end{aligned} \tag{3}$$

where $COM_{i,t}$, $EVN_{i,t}$, $CGOV_{i,t}$, $EMP_{i,t}$, $DIV_{i,t}$, and $PRO_{i,t}$ are the score of CSR's six dimensions, respectively. Since the performance of these dimensions have a mixed effect and will indicate their overall effect on analysts' forecasting, we give equal weight to these six dimensions. Following Huang, Teoh, and Zhang [12], we control firm-level governance and financial variables, including *LISTAGE*, *SIZE*, and *ROA*. YEAR FE and INDUSTRY FE denote year and industry fixed effects, respectively.

3.1.4. Financial Transparency (*OPAQUENESS*)

We use the opaqueness of a company's financial information (*OPAQUENESS*) as a reverse indicator of firms' financial transparency. That is, the higher the opaqueness, the lower the

financial transparency. Following Bhattacharya et al. [24] and Dhaliwal, Radhakrishnan, Tsang, and Yang [5], we employ the following formula to calculate financial transparency:

$$OPAQUENESS_{i,t} = \Big| \frac{\Delta CA_{i,t} - \Delta CL_{i,t} - \Delta CASH_{i,t} + \Delta STD_{i,t} - DEP_{i,t} + \Delta TP_{i,t}}{TA_{i,t-1}} \Big|, \quad (4)$$

where $\Delta CA_{i,t}$ is the change in current assets in the current year, $\Delta CL_{i,t}$ is the change in current liabilities in the current year, and $\Delta CASH_{i,t}$ is the change in cash in the current year. Furthermore, $\Delta STD_{i,t}$ is the change in long-term debt in the current year, $DEP_{i,t}$ represents depreciation and amortization expenses in the current year, $\Delta TP_{i,t}$ is the change in tax expenses in the current year, and $TA_{i,t-1}$ is the total assets in the previous year.

### 3.1.5. Regional Stakeholder-Oriented Culture (*STAKECUL*)

We divide the total amount of charitable donations by the province's gross national product in the current year as the proxy index of the regional stakeholder-oriented culture (*STAKECUL*).

### 3.1.6. Control Variables

Based on studies related to analysts' earnings forecast bias, tone in CSR reports, and non-financial disclosures [1,5,11,21,25–28], we use the following control variables: company size (*SIZE*), the number of years the company has been listed (*LISTAGE*), price-to-book ratio (*BM*), financial leverage (*LEV*), profitability (*ROA*), growth (*GROWTH*), whether it is a state-owned enterprise (*SOE*), the government's shareholding (*SOEPEC*), the holdings of the largest shareholder (*BOPEC*), the tone of the annual report (*LMTONE*), and the number of analysts following the company or analyst focus (*ANAFOCUS*). We also included the number of company-related news reports (*NEWS*), representing the degree of the company's public exposure, and CSR performance (*KLD*), denoting the level of the company's CSR performance. Moreover, star analyst (*STAR*), representing an analyst's forecast level, and analyst experience (*REXP*), denoting the average experience of all analysts following the company, were also included. Finally, we included the relative number of forecasts (*RNUMBER*), denoting the number of forecast reports by an analyst relative to that by other analysts following the company, and the relative accuracy of forecasts (*RACC*), representing an analyst's forecast bias relative to that of other analysts following the company. Table 1 presents the definition of all variables.

**Table 1.** Variable definitions.

| Type of Variable | Variable | Symbol | Definition |
|---|---|---|---|
| Dependent Variables | Analyst earnings forecast bias | BIAS1 | Difference between analyst earnings forecasts and the true value of corporate earnings, divided by average opening price at the beginning of the year |
| | | BIAS2 | Difference between analysts' forecasts of a listed company's EPS and the actual value of the company's EPS divided by the absolute actual value of the company's EPS |
| Independent Variables | Abnormally positive tone in the CSR report | ABTONE | Residual term in the regression model between the tone in the CSR report (TONE) and the actual CSR performance |
| Moderators | Firm financial transparency | OPAQUENESS | Accrual items as a proxy for financial transparency |
| | Stakeholder orientation culture | STAKECUL | Charitable donation divided by the gross national product |

**Table 1.** *Cont.*

| Type of Variable | Variable | Symbol | Definition |
|---|---|---|---|
| Instrumental Variables | Industry average for unusually positive tone | *AV_ABTONE* | Industry average of abnormally positive tone in CSR reports |
| | Regional religion | *RELIGION* | The number of religion sites for the province in which the sample company is registered as a proxy for religion atmosphere |
| Control Variables | CSR performance | *KLD* | CSR performance score from the CNRDS database |
| | Company size | *SIZE* | Natural logarithm of a company's total assets for the current year |
| | Years of company listing | *LISTAGE* | Difference between the current year and the year that the company listed |
| | Price-to-book ratio | *BM* | A company's book value of shareholders' equity for the previous year, divided by the company's market value |
| | Financial leverage | *LEV* | A company's debt-to-assets ratio for the current year |
| Control Variables, con't | Profitability | *ROA* | A company's return on assets for the current year |
| | Growth | *GROWTH* | A company's current operating cash flow divided by the company's total market value for the previous year |
| | Nature of equity | *SOE* | 1 if the equity holder is a state-owned company, and 0 otherwise |
| | Shareholding ratio of state-owned shareholders | *SOEPEC* | Shareholding ratio of a company's state-owned shareholders |
| | Shareholding ratio of the largest shareholder | *BOPEC* | Shareholding ratio of a company's largest shareholder |
| | Analyst focus | *ANAFOCUS* | Number of analysts following a company's analysis for the current year |
| | Report focus | *REPOFOCUS* | Number of research reports following a company's analysis for the current year |
| | Star analyst | *STAR* | Whether the forecast is published by a star analyst or not |
| | Number of company-related news reports | *NEWS* | Number of news reports featuring a company within one day |
| | Tone of annual report | *LMTONE* | Tone of annual report calculated according to Loughran and McDonald [21] |
| | Relative experience | *REXP* | Experience of the jth analyst following company i minus the average experience of all analysts following company i |
| | Relative forecasting number | *RNUMBER* | The average number of companies forecasted by the jth analyst, minus the number of companies forecasted by all analysts following company i |
| Control Variables, con't | Relative forecasting accuracy | *RACC* | The average of the absolute forecasting error of the jth analyst following company i, minus the average of absolute forecasting error of all other analysts following company i, then divided by the average of absolute forecasting error of all other analysts following company i |

### 3.2. Data Source and Sample Selection

Our sample comprised listed Chinese companies that disclosed CSR reports between 2006 and 2016. This is because few listed companies released CSR reports before 2006, and only pre-2016 data could be obtained for CSR performance from our database.

We built a Chinese lexicon for analyzing the tone in CSR reports in the following manner. First, we compared data obtained from the China Stock Market and Accounting Research (CSMAR) Database, the Wind Economic Database, and the CNRDS database to identify those listed companies that issued CSR reports between 2006 and 2016. Subsequently, we downloaded their CSR reports. Second, we randomly selected 300 CSR reports, from which we identified 2619 positive words and 1073 negative words using the dictionaries of Loughran and McDonald [21] and You et al. [22]. We also built a list of stop words based on those from the Harbin Institute of Technology, Baidu Inc., and the Machine Intelligence Lab at Sichuan University, and segmented words using the Jieba Word Segmentation Tool. After removing the stop words, we obtained the final number of positive and negative words in each CSR report using the "Bag of Words" method.

Our lexicon has an advantage over the existing ones that were created using annual reports and management discussions. As stated in Sections 1 and 2.1, companies have a great degree of discretion in creating their CSR reports, and the scope of CSR is broad. Moreover, firms' social responsibility-centric activities differ from other operations. Resultingly, the language in their CSR reports is unique. Therefore, the tone lexicon built from CSR reports may be more suitable for this study.

Data on analysts' earnings forecasts, moderators, instrumental variables, and control variables were collected from the CSMAR Database. We excluded the data of listed companies operating in the financial and insurance industries, and also excluded ST and *ST companies from our sample because they are subject to different regulations. Finally, we winsorize all continuous variables at the 1% level [29–39].

### 3.3. Empirical Model

We use the following regression model for testing Hypothesis 1.

$$BIAS1_{i,t,j} = \alpha_0 + \alpha_1 \cdot ABTONE_{i,t-1} + \gamma \cdot CONTROLS_{i,t-1} + YEAR\ FE + INDUSTRY\ FE + \varepsilon_{i,t}, \qquad (5)$$

where CONTROLS denotes the control variables. We add the interaction variables of abnormally positive tone and financial transparency ($ABTONE \times OPAQUENESS$), and abnormally positive tone and actual CSR performance ($ABTONE \times STAKECUL$) into Equation (5) to test Hypotheses 2 and 3, respectively.

## 4. Results and Discussion

### 4.1. Descriptive Statistics

Table 2 lists the descriptive statistics of all variables. We compiled 140,832 analyst earnings forecasts, with a forecast bias of 0.0199, on average, and a standard deviation of 0.0326. In the full sample, the average value of abnormal tone in CSR reports is 0.0015, with a maximum of 0.5205 and a minimum of −0.2347. Therefore, significant differences exist in the degree of firms' tone management in CSR reports, with most companies using an "inflated" tone. *KLD* is 17.1190 on average, with a variance of 5.6508; the difference between maximum and minimum values is 35. Thus, significant differences exist among CSR performance of Chinese-listed companies.

The index for financial transparency is 0.0931, on average, and the difference between its maximum and minimum values is 0.5655. This result indicates that companies generally exhibit a respectable level of transparency. *STAKECUL* is 0.0008 on average, and its standard deviation is 0.0007. Since Chinese capital markets are yet to mature, there are incomplete laws and regulations, and listed companies with good or poor corporate governance coexist in the market, the control variables reflect the varied and uneven statistical characteristics of the sample companies.

**Table 2.** Descriptive statistics.

| Variable | Obs. | Mean | Std. Dev. | Min | Max |
|---|---|---|---|---|---|
| *BIAS1* | 140,832 | 0.0199 | 0.0326 | −0.0422 | 0.1898 |
| *ABTONE* | 3075 | 0.0015 | 0.0814 | −0.2374 | 0.5205 |
| *OPAQUENESS* | 3004 | 0.0931 | 0.0815 | 0.0001 | 0.5656 |
| *STAKECUL* | 3075 | 0.0008 | 0.0007 | 0.00001 | 0.0053 |
| *KLD* | 3075 | 17.1190 | 5.6508 | 2 | 37 |
| *SIZE* | 3075 | 23.1147 | 1.3620 | 18.2659 | 28.0554 |
| *LISTAGE* | 3075 | 2.4399 | 0.6527 | 0.0000 | 3.5553 |
| *BM* | 3075 | 0.5220 | 0.3334 | 0.0992 | 1.5447 |
| *LEV* | 3075 | 0.4819 | 0.1861 | 0.0453 | 0.8804 |
| *ROA* | 3075 | 0.0523 | 0.0501 | −0.2248 | 0.3317 |
| *GROWTH* | 3075 | 0.0660 | 0.1149 | −1.0229 | 1.0262 |
| *SOE* | 3075 | 0.5932 | 0.4913 | 0 | 1 |
| *SOEPEC* | 3075 | 0.0401 | 0.1075 | 0.0000 | 0.6136 |
| *BOPEC* | 3075 | 0.3738 | 0.1511 | 0.0639 | 0.7498 |
| *ANAFOCUS* | 3075 | 2.2585 | 0.9764 | 0.0000 | 4.3307 |
| *NEWS* | 3075 | 5.7765 | 1.0799 | 0.6931 | 10.8083 |
| *LMTONE* | 3075 | 0.0090 | 0.0734 | −0.2425 | 0.3391 |
| *STAR* | 3075 | 0.1932 | 0.3949 | 0 | 1 |
| *REXP* | 140,832 | 0.7068 | 7.8750 | −13.5200 | 26.0000 |
| *RNUMBER* | 140,832 | −0.3529 | 24.2015 | −59.8000 | 160.6500 |
| *RACC* | 140,832 | −2.0167 | 0.7887 | −6 | 1 |
| *AV_ABTONE* | 242 | 0.0045 | 0.0311 | −0.0967 | 0.1829 |
| *RELIGION* | 3075 | 14.5369 | 9.7204 | 1 | 31 |

*4.2. Correlation Analysis*

Table 3 shows the correlation coefficients of the variables. The coefficient of *BIAS1* with *ABTONE* is 0.015, significantly positive at the 1% level, aligning with H1. This result provides preliminary evidence that the tone in CSR reports has a certain impact on analysts' forecasts, and that an abnormally positive tone can induce a certain level of optimistic bias in the forecasts.

**Table 3.** Pearson correlation coefficients.

| | *BIAS1* | *TONE* | *ABTONE* | *KLD* | *OPAQUENESS* | *STAKECUL* |
|---|---|---|---|---|---|---|
| *BIAS1* | 1.000 | | | | | |
| *TONE* | 0.022 *** | 1.000 | | | | |
| *ABTONE* | 0.015 *** | 0.826 *** | 1.000 | | | |
| *KLD* | −0.098 *** | −0.092 *** | −0.014 *** | 1.000 | | |
| *OPAQUENESS* | 0.032 *** | −0.072 *** | 0.060 *** | −0.075 *** | 1.000 | |
| *STAKECUL* | −0.114 *** | −0.073 *** | −0.025 *** | 0.142 *** | −0.055 *** | 1.000 |

Note: This table reports the Pearson correlation matrix of the main variables. Variable definitions are provided in Table 1. *, **, and *** denote significance at the 10%, 5%, and 1% level, respectively.

The coefficient of *KLD* with *BIAS1* is −0.098 and significantly negative. This result provides preliminary evidence that analysts' forecasts of companies with poor CSR performance tend to exhibit an optimistic bias. This phenomenon can be explained based on

the stakeholder theory and legitimacy theory. Companies with poor CSR performance are motivated to manage the tone in their CSR reports, which can mislead analysts.

We calculated the variance inflation factor (VIF) for the regression models. The VIF of all models is less than 8, and the average VIF is 3.18. Therefore, multicollinearity is not an issue for the regression models [18–20].

### 4.3. Regression Analysis

Based on the descriptive statistics and correlation analysis, we perform regression analysis using the models in Section 3.

### 4.3.1. Impact of the Tone in CSR Reports on the Optimistic Bias in Analysts' Forecasts

Table 4 shows the results regarding the impact of abnormally positive tone in CSR reports on the optimistic bias in analysts' forecasts, obtained by employing Equation (5). Column (1) reports the results, excluding the control variables. Abnormally positive tone significantly and positively affects analysts' earnings forecast bias at the 1% level. Column (2) presents the results incorporating the control variables. The coefficient of *ABTONE* is 0.0038. Again, abnormally positive tone significantly and positively affects analysts' earnings forecast bias at the 1% level.

**Table 4.** Effects of abnormal positive tone in CSR reports on analyst forecast bias Pearson correlation coefficients.

| | (1) $BIAS1_{i,t,j}$ | (2) $BIAS1_{i,t,j}$ |
|---|---|---|
| $ABTONE_{i,t-1}$ | 0.0053 *** (4.69) | 0.0038 *** (3.46) |
| $SIZE_{i,t-1}$ | | 0.0033 *** (22.71) |
| $LISTAGE_{i,t-1}$ | | −0.0013 *** (−7.98) |
| $BM_{i,t-1}$ | | 0.0007 *** (2.84) |
| $LEV_{i,t-1}$ | | −0.0055 *** (−26.01) |
| $ROA_{i,t-1}$ | | 0.0109 *** (65.39) |
| $GROWTH_{i,t-1}$ | | −0.001 *** (−13.78) |
| $SOE_{i,t-1}$ | | −0.0054 *** (−24.14) |
| $SOEPEC_{i,t-1}$ | | −0.004 *** (−4.02) |
| $BOPEC_{i,t-1}$ | | −0.0004 ** (−2.47) |
| $ANAFOCUS_{i,t-1}$ | | −0.0027 *** (−20.25) |
| $NEWS_{i,t-1}$ | | 0.0008 *** (5.17) |
| $LMTONE_{i,t-1}$ | | 0.0043 *** (32.77) |

**Table 4.** *Cont.*

| | (1) $BIAS1_{i,t,j}$ | (2) $BIAS1_{i,t,j}$ |
|---|---|---|
| $KLD_{i,t-1}$ | | −0.0004 *** (−22.51) |
| $STAR_{i,t-1,j}$ | | −0.0007 *** (−3.61) |
| $REXP_{i,t-1,j}$ | | −0.0007 *** (−6.08) |
| $RNUMBER_{i,t-1,j}$ | | 0.0001 *** (4.85) |
| $RACC_{i,t-1,j}$ | | −0.0005 ** (−2.32) |
| YEAR FE | YES | YES |
| INDUSTRY FE | YES | YES |
| ADJ-R$^2$ | 0.0880 | 0.1595 |
| N | 140,832 | 140,832 |

Note: This table estimates the effect of abnormally positive tone in CSR reports on analyst forecast bias. Variable definitions are provided in Table 1. T statistics are reported in parentheses. *, **, and *** indicate significance at the 10%, 5%, and 1% levels (two-tailed), respectively.

One standard deviation in *ABTONE* will increase the forecast bias 1.55%, relative to its mean. For comparison (1.554% = 100% × 0.0814 × 0.0038 ÷ 0.0199), Huang, Teoh, and Zhang [12] found that one standard deviation increase in *ABTONE* corresponds to a decrease of 0.20%, relative to the median earnings. We note the following differences: instead of the level value of earnings, we focus on forecast accuracy; furthermore, our sample focuses on Chinese firms, whereas the previous study relied on Compustat. Based on these results, an abnormal tone with a positive emotional aspect will increase the persuasiveness of CSR reports and public confidence, inducing analysts to judge company value as favorable and creating an optimistic bias. Therefore, H1 is supported.

4.3.2. Moderating Effects

In this subsection, we examine the impact of two important moderators: a firm's financial transparency and regional culture, regarding the relationship between abnormal CSR tone and analyst forecasting.

Columns (1) and (2) in Table 5 report the regression results after testing H2. The coefficients of the interaction term *ABTONE × OPAQUENESS* are significant and positive. This result suggests that a firm's financial opaqueness strengthens the relationship between abnormal positive tone in CSR reports and analysts' earnings forecasting bias, supporting H2.

**Table 5.** Moderating effects.

| | (1) $BIAS1_{i,t,j}$ | (2) $BIAS1_{i,t,j}$ | (3) $BIAS1_{i,t,j}$ | (4) $BIAS1_{i,t,j}$ |
|---|---|---|---|---|
| $ABTONE_{i,t-1}$ | 0.0106 *** (3.28) | 0.0146 *** (4.69) | 0.0257 *** (11.27) | 0.0181 *** (7.94) |
| $OPAQUENESS_{i,t-1}$ | 0.0006 *** (7.99) | 0.0009 *** (11.90) | | |
| $ABTONE_{i,t-1} \times OPAQUENESS_{i,t-1}$ | 0.0015 (1.43) | 0.0034 *** (3.25) | | |

**Table 5.** *Cont.*

| | (1) $BIAS1_{i,t,j}$ | (2) $BIAS1_{i,t,j}$ | (3) $BIAS1_{i,t,j}$ | (4) $BIAS1_{i,t,j}$ |
|---|---|---|---|---|
| $STAKECUL_{i,t-1}$ | | | 2.1642 *** (13.52) | 1.5848 *** (9.48) |
| $ABTONE_{i,t-1} \times STAKECUL_{i,t-1}$ | | | 16.6525 *** (7.67) | 8.0401 *** (3.67) |
| $SIZE_{i,t-1}$ | | 0.0035 *** (23.72) | | 0.003 *** (15.52) |
| $LISTAGE_{i,t-1}$ | | −0.0013 *** (−7.61) | | −0.0011 *** (−5.48) |
| $BM_{i,t-1}$ | | 0.0006 ** (2.49) | | 0.0001 (0.28) |
| $LEV_{i,t-1}$ | | −0.0059 *** (−27.60) | | −0.0073 *** (−24.51) |
| $ROA_{i,t-1}$ | | 0.011 *** (65.45) | | −0.0134 *** (−57.49) |
| $GROWTH_{i,t-1}$ | | −0.001 *** (−12.79) | | −0.0007 *** (−7.46) |
| $SOE_{i,t-1}$ | | −0.0053 *** (−23.48) | | −0.006 *** (−19.8) |
| $SOEPEC_{i,t-1}$ | | −0.0029 *** (−2.87) | | 0.0025 ** (2.07) |
| $BOPEC_{i,t-1}$ | | −0.0006 *** (−3.66) | | 0.0011 *** (5.38) |
| $ANAFOCUS_{i,t-1}$ | | −0.0027 *** (−20.04) | | −0.0052 *** (−23.96) |
| $NEWS_{i,t-1}$ | | 0.0006 *** (3.75) | | 0.0025 *** (10.76) |
| $LMTONE_{i,t-1}$ | | 0.0045 *** (34.12) | | −0.0057 *** (−33.61) |
| $KLD_{i,t-1}$ | | −0.0004 *** (−21.45) | | −0.0005 *** (−21.16) |
| $STAR_{i,t-1,j}$ | | −0.0006 *** (−3.35) | | −0.0006 *** (−2.66) |
| $REXP_{i,t-1,j}$ | | −0.0006 *** (−6.11) | | −0.0001 *** (−5.28) |
| $RNUMBER_{i,t-1,j}$ | | 0.0013 *** (4.87) | | 0.0000 *** (4.12) |
| $RACC_{i,t-1,j}$ | | −0.0004 ** (−2.25) | | −0.0001 ** (−2.17) |
| YEAR FE | YES | YES | YES | YES |
| INDUSTRY FE | YES | YES | YES | YES |
| ADJ-$R^2$ | 0.0873 | 0.1598 | 0.1005 | 0.1976 |
| N | 138,434 | 138,434 | 140,832 | 140,832 |

Note: This table estimates the moderating effects of financial transparency and local stakeholder orientation culture. Variable definitions are provided in Table 1. T statistics are reported in parentheses. *, **, and *** indicate significance at the 10%, 5%, and 1% levels (two-tailed), respectively.

Next, we consider the regional culture. Columns (3) and (4) in Table 5 present the regression results after testing H3. In Column (4), the coefficient of the interaction term

is 8.0401, and significant at the 1% level. This result indicates that a stakeholder-centric culture amplifies the impact of abnormal positive tone in CSR reports on analysts' earnings forecast bias, supporting H3. That is, the tone in CSR reports becomes more powerful when the regional culture is highly stakeholder-oriented.

## 5. Robustness Check

### 5.1. Addressing Potential Endogeneity Problems

We address potential endogeneity problems using two instrumental variables: industry average abnormally positive tone in CSR reports (*AV_ABTONE*) and regional religion (*RELIGION*).

*AV_ABTONE* is the average abnormal positive tone in the CSR reports of other companies operating within the same industry and year. Different industries have different production and operation methods and stakeholders, and thus, companies in different industries will have different tones in their CSR reports. Since firms and their industry peers face similar industrial regulations, legal norms, and business environment, the abnormal positive tone in a firm's CSR reports is likely to relate to the industry average. This industry average tone should be considered an industry characteristic that should not induce a bias in analysts' earnings forecast. Hence, the variable is exogenous.

Column (1) in Table 6 presents the results of the first-stage regression. The coefficient on the instrumental variable (*AV_ABTONE*) is significantly positive, indicating that the higher the industry average, the more positive the abnormal tone in CSR reports. Column (2) shows the results of the second-stage regression, indicating that the predicted abnormal positive tone in CSR reports positively and significantly relates to analysts' earnings forecast bias, even when the industry's average abnormally positive tone is used as the instrumental variable.

Following Deng et al. [23] and Muslu et al. [14], we selected another instrumental variable: the religious culture of each province or regional religion. Owing to China's vast size, obvious cultural differences exist across different regions. Often, religion differences produce different values and approaches. Companies operating in areas where the atmosphere is highly religious generally prefer fewer risks and more prudent investments. Hence, firms located in highly religious provinces would emphasize selecting the tone in CSR reports since CSR investments have an insurance effect.

Since regional culture is passed down through generations, it is an exogenous variable for companies. This variable influences abnormally positive tone in CSR reports, but has little effect on analysts' forecasts. Sourcing information from the National Bureau of Statistics of China, we calculate the regional religious culture (*RELIGION*) as the sum of the total number of religious venues in the province (including Buddhist temples, Taoist temples, mosques, Catholic churches, and Christian churches) [23]. The higher the number of religious venues, the higher the index of religious culture in the province.

Column (3) in Table 6 presents the results of the first-stage regression after substituting the instrumental variable *RELIGION* into the 2SLS model. The coefficient of the instrument variable (*RELIGION*) is significantly negative, indicating that companies operating in highly religious regions do not actively use an abnormal positive tone in CSR disclosures. This result arises owing to the estranged relationship between companies and stakeholders in highly religious regions. This estrangement increases the focus on CSR and results in a possible lack of opportunism around CSR disclosures. Firms operating in highly religious regions are greatly constrained by stakeholders, and thus, they emphasize socially responsible behavior to stay close to the stakeholders. Consequently, they demonstrate a relatively respectable CSR performance and are more cautious in their actual disclosures. Column (4) displays the results of the second-stage regression. The predicted abnormal positive tone significantly and positively relates to analysts' earnings forecast bias, even when regional culture is used as the instrument variable.

**Table 6.** Instrumental variable approach.

| | (1) First Stage $ABTONE_{i,t-1}$ | (2) Second Stage $BIAS1_{i,t,j}$ | (3) First Stage $ABTONE_{i,t-1}$ | (4) Second Stage $BIAS1_{i,t,j}$ |
|---|---|---|---|---|
| $AV\_ABTONE_{i,t-1}$ | 1.0141 *** (112.0094) | | | |
| $RELIGION_{i,t-1}$ | | | −0.0003 *** (−14.3815) | |
| $Predicted\ ABTONE_{i,t-1}$ | | 0.0316 *** (8.2382) | | 0.5818 *** (12.0000) |
| $SIZE_{i,t-1}$ | 0.0019 *** (5.7423) | 0.0033 *** (22.6645) | 0.0009** (2.5268) | 0.0032 *** (12.7804) |
| $LISTAGE_{i,t-1}$ | 0.0001 (0.0773) | −0.0013 *** (−8.2220) | −0.001 *** (−2.5974) | −0.0006 ** (−2.0289) |
| $BM_{i,t-1}$ | −0.002 *** (−3.4390) | 0.0006 *** (2.5600) | −0.0023 *** (−3.7515) | 0.0018 *** (4.1251) |
| $LEV_{i,t-1}$ | −0.0003 (−0.5337) | −0.0055 *** (−25.8679) | −0.0002 (−0.4641) | −0.0056 *** (−15.5204) |
| $ROA_{i,t-1}$ | −0.0024 *** (−6.1291) | −0.0109 *** (−65.3976) | −0.0019 *** (−4.5559) | −0.0101 *** (−34.5199) |
| $GROWTH_{i,t-1}$ | 0.0015 *** (8.7109) | −0.001 *** (−13.2296) | 0.001 *** (5.7569) | −0.0016 *** (−11.7378) |
| $SOE_{i,t-1}$ | −0.0019 *** (−3.7454) | −0.0055 *** (−24.4560) | −0.0021 *** (−3.8358) | −0.0038 *** (−9.4410) |
| $SOEPEC_{i,t-1}$ | 0.0206 *** (8.8540) | −0.0035 *** (−3.4944) | 0.0165 *** (6.8021) | −0.0123 *** (−6.6234) |
| $BOPEC_{i,t-1}$ | 0.0016 *** (4.3910) | −0.0003** (−2.1555) | 0.0017 *** (4.3328) | −0.0012 *** (−4.1564) |
| $ANAFOCUS_{i,t-1}$ | 0.0038 *** (12.0848) | −0.0027 *** (−19.8952) | 0.0010 *** (3.1038) | −0.0033 *** (−13.9933) |
| $NEWS_{i,t-1}$ | −0.0022 *** (−5.7199) | 0.0009 *** (5.2978) | 0.0004 (1.0192) | 0.0004 (1.5676) |
| $LMTONE_{i,t-1}$ | −0.0038 *** (−12.3133) | −0.0045 *** (−33.6498) | −0.0045 *** (−14.2333) | −0.0015 *** (−4.5995) |
| $KLD_{i,t-1}$ | −0.0003 *** (−7.2212) | −0.0004 *** (−22.5235) | −0.0001 * (−1.8926) | −0.0004 *** (−12.1262) |
| $STAR_{i,t-1}$ | −0.0002 (−0.4080) | −0.0007 *** (−3.6223) | −0.0002 (−0.4226) | −0.0006* (−1.8109) |
| $REXP_{i,t-1,j}$ | 0.0001 ** (2.21) | −0.0009 *** (−5.64) | 0.0001 ** (1.98) | −0.0002 *** (−5.56) |
| $RNUMBER_{i,t-1,j}$ | 0.0009 ** (−2.56) | 0.0002 *** (3.70) | −0.0002 *** (−2.99) | 0.0001 *** (4.73) |
| $RACC_{i,t-1,j}$ | 0.0002 ** (2.21) | 0.0008 * (−1.68) | 0.0001** (2.01) | −0.0001 *** (−2.92) |
| YEAR FE | YES | YES | YES | YES |
| INDUSTRY FE | YES | YES | YES | YES |
| F-VALUE | 296.15 | | 31.29 | |
| ADJ-R$^2$ | 0.0865 | 0.1536 | 0.0129 | 0.0439 |
| N | 140,832 | 140,832 | 140,832 | 140,832 |

Note: This table presents the results with instrumental variables. Variable definitions are provided in Table 1. T statistics are reported in parentheses. *, **, and *** indicate significance at the 10%, 5%, and 1% levels (two-tailed), respectively.

### 5.2. Using an Alternative Measure of Analysts' Forecast Bias

To further verify that tone can influence analysts' forecast bias, based on Becchetti et al. [7], we examine earnings forecast bias using a relative value indicator (*BIAS2*). This is calculated as the relative value of the difference between analysts' forecasts of a listed company's

EPS and the actual value of the company's EPS divided by the absolute actual value of the company's EPS, as shown below:

$$BIAS2_{i,t,j} = \frac{Feps_{i,t,j} - Aeps_{i,t}}{|Aeps_{i,t}|}, \tag{6}$$

where $Feps_{i,t,j}$ is analysts' forecast of the company's EPS for the current year following the day the company's CSR report is published. $Aeps_{i,t}$ is the actual value of the company's EPS for the current year.

　　　Table 7 shows the results of the regression analysis when the alternative measure, *BIAS2*, is employed in Equation (5). Column (1) displays the positive impact of abnormal positive tone on analysts' earnings forecast bias. Columns (2) and (3) incorporate the moderators. The coefficients of the interaction terms *ABTONE × OPAQUENESS* and *ABTONE × STAKECUL* are significant and positive. Hence, our results remain unchanged, even when an alternative measure of analysts' forecast bias is used.

**Table 7.** Substituting an alternative measure of analyst forecast bias.

| | (1) $BIAS2_{i,t,j}$ | (2) $BIAS2_{i,t,j}$ | (3) $BIAS2_{i,t,j}$ |
|---|---|---|---|
| $ABTONE_{i,t-1}$ | 0.8430 *** (8.03) | 2.8707 *** (9.65) | 3.4982 *** (6.03) |
| $OPAQUENESS_{i,t-1}$ | | 0.0709 *** (10.23) | |
| $ABTONE_{i,t-1} \times OPAQUENESS_{i,t-1}$ | | 0.6601 *** (6.66) | |
| $STAKECUL_{i,t-1}$ | | | 1.5973 *** (5.09) |
| $ABTONE_{i,t-1} \times STAKECUL_{i,t-1}$ | | | 0.2770 *** (9.15) |
| CONTROLS$_{i,t-1}$ | YES | YES | YES |
| YEAR FE | YES | YES | YES |
| INDUSTRY FE | YES | YES | YES |
| ADJ-R$^2$ | 0.1581 | 0.1596 | 0.1970 |
| N | 140,832 | 138,434 | 140,832 |

Note: This table presents the results with an alternative measure of analyst forecasting bias. Variable definitions are provided in Table 1. T statistics are reported in parentheses. *, **, and *** indicate significance at the 10%, 5%, and 1% levels (two-tailed), respectively.

### 5.3. Replacing the Independent Variable with the Change in Abnormal Tone (ΔABTONE)

　　　Following Huang, Teoh, and Zhang [12], we use the difference between abnormal positive tone in the current and previous CSR reports (*ΔABTONE*), i.e., the change in abnormal positive tone, to test the robustness of our results.

　　　There are two potential shortcomings to using level of tone to measure the text's emotional tendency [15]. First, managers tend to refer to the text in previous reports when crafting the text in the current report. Hence, the current report may reflect the tone used previously. Second, the measurement of tonal tendency depends on the lexicon selected for research, but that lexicon may fail to cover the emotional tendencies of the vocabulary in specific industries, companies, and events; it takes time for a lexicon to cover such specificities. Thus, a change in tone can reflect an increase in the emotional tendency of text. Considering this possibility, we apply the change in abnormal positive tone in our regression analysis.

Column (1) in Table 8 shows the results of regression analysis when $\Delta ABTONE$ is employed into Equation (5). In Column (1), the coefficient of $\Delta ABTONE$ with analysts' earnings forecast bias is 0.0052 and significant at the 1% level.

**Table 8.** Using alternative abnormal tone measures.

| | (1) $BIAS1_{i,t,j}$ | (2) $BIAS1_{i,t,j}$ |
|---|---|---|
| $\Delta ABTONE_{i,t-1}$ | 0.0052 *** (3.89) | |
| $ABTONE\_LM_{i,t-1}$ | | 0.0011 *** (3.98) |
| $CONTROLS_{i,t-1}$ | YES | YES |
| YEAR FE | YES | YES |
| INDUSTRY FE | YES | YES |
| ADJ-R$^2$ | 0.1591 | 0.1586 |
| N | 109,094 | 140,832 |

Note: This table presents the results with alternative measures of abnormally positive tone in CSR reports. $\Delta ABTONE$ is the difference between abnormal positive tone in the current and previous CSR reports. $ABTONE\_LM$ is the abnormal positive tone calculated using the Chinese version of the L and M lexicon. Variable definitions are provided in Table 1. T statistics are reported in parentheses. *, **, and *** indicate significance at the 10%, 5%, and 1% levels (two-tailed), respectively.

*5.4. Using the Abnormal Tone Sourced from another Tone Lexicon*

The CNRDS provides the Loughran and McDonald (or L and M) tone dictionary list, based on the financial sentiment English words that Loughran and McDonald [21] provided. The CNRDS translates the list into Chinese and removes some words not commonly used in the Chinese language. Finally, the Chinese version of the L and M dictionary list contained 1076 positive words and 2080 negative words. We obtained the abnormal positive tone index ($ABTONE\_LM$) using this Chinese version of the L and M tone dictionary list.

Column (2) in Table 8 shows regression results when the primary explanatory variable is $ABTONE\_LM$. The coefficient of this new abnormal positive tone is 0.0011 and significant at the level of 1%. Considering the coefficient's direction and significance level, our results remain robust.

## 6. Mechanism through which tone Management in CSR Reports Generates Analyst Forecast Bias

Regarding why and how an abnormal tone in CSR reports affects the accuracy of analysts' forecasts, we posit that firms with poor CSR performance manipulate the tone of CSR disclosure to address the pressures of outsiders, which misdirects analysts. To support our argument, we test whether the relationship between abnormal positive tone and analysts' earnings forecast bias is stronger when the CSR performance is poor.

Table 9 presents the results. Columns (2) shows that the coefficient on the interaction of abnormal positive tone and actual CSR performance, $ABTONE \times KLD$, is significantly negative. This result indicates that a company demonstrating poor CSR performance is more likely to manage tone in its CSR report, hoping that the abnormally positive tone induces an optimistic bias in analysts' earnings forecasts. The behavior of managing tone demonstrates a company's intention to conceal the shortcomings in its actual CSR performance and create a positive social impression.

**Table 9.** Mechanism analysis.

| | (1) Without Interaction Term $BIAS1_{i,t,j}$ | (2) With Interaction Term $BIAS1_{i,t,j}$ | (3) ABTONE and KLD Are Centered $BIAS1_{i,t,j}$ |
|---|---|---|---|
| $ABTONE_{i,t-1}$ | 0.0039 *** (4.55) | 0.0491 *** (13.64) | 0.0042 *** (5.02) |
| $ABTONE_{i,t-1} \times KLD_{i,t-1}$ | | −0.0026 *** (−13.21) | −0.0032 *** (−20.40) |
| $KLD_{i,t-1}$ | −0.0005 *** (−35.49) | −0.0003 *** (−20.13) | −0.0005 *** (−32.18) |
| CONTROLS$_{i,t-1}$ | YES | YES | YES |
| YEAR FE | YES | YES | YES |
| INDUSTRY FE | YES | YES | YES |
| ADJ-R$^2$ | 0.2595 | 0.1606 | 0.2616 |
| N | 140,832 | 140,832 | 140,832 |

Note: This table presents the results of testing whether the relationship between abnormal positive tone and analysts' earnings forecast bias is stronger when the CSR performance is poor. Variable definitions are provided in Table 1. T statistics are reported in parentheses. *, **, and *** indicate significance at the 10%, 5%, and 1% levels (two-tailed), respectively.

We used mean centering of indicators for each first-order term to address the correlations among first-order and interaction factors, and associated multicollinearity problems [38]. We modified Table 9 and some texts in the revised manuscript accordingly. Column (3) of Table 9 reports the results, which show that our relevant conclusion is unchanged. In addition, we use the results of the regression analysis without interaction term (Column (1) of Table 9) for comparison.

## 7. Conclusions

In the field of social responsibility, it is continually debated how a company's CSR disclosure affects the market's expectations of company value. Based on the findings of this study, the textual tone in CSR reports, which is a crucial form of non-financial disclosure, significantly affects how the market, as represented by analysts, judges a company's value. This study concluded that an abnormally positive tone in CSR reports will induce an optimistic bias in analysts' earnings forecast. This phenomenon is strikingly prominent in companies with poor financial transparency and those operating in regions with a stakeholder-oriented culture. Moreover, the relationship between abnormal positive tone and analysts' earnings forecast bias is stronger when the CSR performance is poor, providing further evidence that firms use abnormal tone opportunistically.

The non-quantitative information stemming from an abnormal positive tone can significantly boost market confidence. Companies are highly motivated to manage the tone in their CSR report in China's yet-to-mature CSR fulfillment and disclosure system. Therefore, it becomes imperative to establish a strict CSR disclosure and review system, with the expression of the text in disclosures being regulated. Moreover, standards must be set regarding text characteristics, which should suit the field of social responsibility. This action will provide rules that regulatory authorities and the market could utilize. External audit organizations should also assist with auditing the authenticity and textual integrity of CSR reports, helping regulatory authorities and investors in judging the compliance and reliability of CSR reports.

This study may be useful for companies and their managers. A company's non-financial disclosure and the market's judgment of its value constitute a circular process. If the company intends to manipulate CSR reports in a short-sighted manner, sending signals using CSR disclosure will be ineffective in the long run.

A pertinent question for empirical studies is always whether the empirical relationships identified in the study could change when one uses alternative datasets. Future research can improve this by adding the latest data or conducting the study internationally. Another question that must be addressed in future research is whether the analysts could find and respond to the abnormal tone.

Our results are based on Chinese listed firms. Due to the substantial institutional differences between China and other emerging countries, our results may not be generalizable to other markets.

**Author Contributions:** Conceptualization, H.W. and X.L.; methodology, H.W. and X.L.; software, X.L.; validation, H.W.; formal analysis, X.L.; data curation, X.L.; writing—original draft preparation, H.W. and X.L.; writing—review and editing, H.W.; supervision, H.W.; funding acquisition, H.W. All authors have read and agreed to the published version of the manuscript.

**Funding:** This research was funded by the National Social Science Fund: Study on the Motivation, Mechanism and Governance Effect of Enterprises' Carbon Greenwashing (No. 22BGL081).

**Conflicts of Interest:** The authors declare no conflict of interest.

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
