# Peer review of "Does the Tone in Corporate Social Responsibility Reports Misdirect Analysts’ Forecasts in China?"

_sustainability, doi:10.3390/su142416631_

Round 1
Reviewer 1 Report
The manuscript deals with an interesting and actual topic and aligns with the journal's aims. The abstract is well-written; however, the originality of the study should be emphasized more. The structure of the survey is acceptable. It should be noted that a comprehensive and systematic literature review on the topic is missing; only a superficial overview has been made related to hypothesis development. The methodology used by the authors fits the research questions. The data source, sample selection, regression model, and variables are well-defined. However, the discussion of the results needs further development; it is recommended to compare own results to previous literature findings to highlight the own contribution to the topic. The limitations of the results and the potential applications of the key findings should be defined in a more detailed and structured way. The article is adequately referenced. Regarding the style and language of the study, a moderate stylistic and language check is needed.
Reviewer 2 Report
Dear Authors,
The author needs to clarify the new contribution of the research in the introduction. It is necessary to clearly state the new and motivating points of the article.
The literature review should be placed after the missing section. Authors need to update recent studies. And point out the missing point to carry out this study. The author should have a literature review to compare the results of previous studies conducted in the same research context.
The author needs to make a clear hypothesis for each pair of moderator variables.
The research hypothesis has not yet highlighted the background theory and arguments for the proposed hypothesis.
The author needs to supplement the model representing the interaction variable with the multicollinearity control method according to Baron and Kenny (1986).
I hope my comments may help you in developing the paper.
Reviewer 3 Report
Dear authors,
Thank you very much for sending your paper to the journal, the following issues should be fixed in the next version:
1-The paper is employed out of date data which should be up to date until 2021
2-the theoretical issues should be improve by the recent studies, some are listed below:
-Do corporate volunteering programs and perceptions of corporate morality impact perceived employer attractiveness?
-The Effect of Corporate Governance Structure on Fraud and Money Laundering
-Does corporate social responsibility yield financial returns in Islamic banking?
-The Impact of International Financial Reporting Standards on Financial Reporting Quality: Evidence from Iraq
-Corporate social responsibility and future financial performance: evidence from Tehran Stock Exchange
3- The research question is not well addressed in the study
4-The research gap is not stated in the paper
5- What is the originality of the paper compared to the published papers?
6- The conclusion does not fully address the research question
Round 2
Reviewer 3 Report
Dear authors,
Thank you very much for sending revised manuscript, however, still the following issues should be addressed in the next version:
1-Why you employed somewhat out of date data in the paper?
2-The theoretical issues should be improve by the most recent studies some are listed below:
-The relationship between managerial entrenchment and firm risk‐taking on social responsibility disclosure
The relationship between corporate governance and financial reporting transparency
